# Pre-Clinical Evaluation of the Antiviral Activity of Epigalocatechin-3-Gallate, a Component of Green Tea, against Influenza A(H1N1)pdm Viruses

**DOI:** 10.3390/v15122447

**Published:** 2023-12-16

**Authors:** Harry Stannard, Paulina Koszalka, Nikita Deshpande, Yves Desjardins, Mariana Baz

**Affiliations:** 1WHO Collaborating Centre for Reference and Research on Influenza, at the Peter Doherty Institute for Infection and Immunity, Melbourne, VIC 3000, Australia; harry.stannard@influenzacentre.org (H.S.);; 2Institute of Nutrition and Functional Foods, Centre Nutrition, Santé et Societé (NUTRISS) Center, Faculté de Sciences de L’agriculture et de L’alimentation (FSAA), Université Laval, Quebec City, QC G1V 4L3, Canada; 3Department of Microbiology and Immunology, at the Peter Doherty Institute for Infection and Immunity, The University of Melbourne, Melbourne, VIC 3000, Australia

**Keywords:** Epigalocatechin-3-gallate, polyphenol, oseltamivir, influenza antiviral treatment, combinational therapy, influenza viruses, ferret model

## Abstract

Influenza antiviral drugs are important tools in our fight against both annual influenza epidemics and pandemics. Polyphenols are a group of compounds found in plants, some of which have demonstrated promising antiviral activity. Previous in vitro and mouse studies have outlined the anti-influenza virus effectiveness of the polyphenol epigallocatechin-3-gallate (EGCG); however, no study has utilised the ferret model, which is considered the gold-standard for influenza antiviral studies. This study aimed to explore the antiviral efficacy of EGCG in vitro and in ferrets. We first performed studies in Madin-Darby Canine Kidney (MDCK) and human lung carcinoma (Calu-3) cells, which demonstrated antiviral activity. In MDCK cells, we observed a selective index (SI, CC50/IC50) of 77 (290 µM/3.8 µM) and 96 (290 µM/3.0 µM) against A/California/07/2009 and A/Victoria/2570/2019 (H1N1)pdm09 influenza virus, respectively. Calu-3 cells demonstrated a SI of 16 (420 µM/26 µM) and 18 (420 µM/24 µM). Ferrets infected with A/California/07/2009 influenza virus and treated with EGCG (500 mg/kg/day for 4 days) had no change in respiratory tissue viral titres, in contrast to oseltamivir treatment, which significantly reduced viral load in the lungs of treated animals. Therefore, we demonstrated that although EGCG showed antiviral activity in vitro against influenza viruses, the drug failed to impair viral replication in the respiratory tract of ferrets.

## 1. Introduction

Influenza virus infections cause a significant global burden on human health. Seasonal influenza epidemics may cause up to 1 billion infections and 650,000 deaths worldwide each year [1,2]. In combination with seasonal influenza vaccines, anti-influenza virus drugs are an important tool to reduce the disease severity of infected patients. The antiviral drugs currently available include the four neuraminidase inhibitors (NAIs), oseltamivir (OST), zanamivir, laninamivir, and peramivir. Additionally, baloxavir marboxil, a polymerase acidic (PA) endonuclease inhibitor, which was first licenced in 2018 in Japan and the USA, is now available in many countries [3]. All licenced antiviral drugs have known resistance mutations that can occur after treatment, and therefore drugs with novel mechanisms of action are important to develop and evaluate. Broad-spectrum compounds that demonstrate anti-influenza properties may provide alternative treatment options compared to currently licenced antiviral drugs. Drugs with novel mechanisms of action are particularly important in the event of widespread viral resistance to licensed antivirals and also provide an opportunity for combination therapy with the current standard of care (SOC) influenza treatment, oseltamivir [4].

Polyphenols are ubiquitous in the plant kingdom and are notably found in grains, vegetables, fruits, spices, and tea leaves [5]. Green tea (*Camellia sinensis* (L.) Kuntze) is one of the most consumed beverages in the world. The leaves are particularly rich in flavan-3-ols such as epigallocatechin (EGC) and epigallocatechin-3-gallate (EGCG). Green tea extract-derived EGCG (and other polyphenols) are proposed to reduce oxidative stress associated with many diseases. EGCG has been researched for its potential therapeutic benefits in treating numerous chronic conditions such as inflammatory disorders, neurodegenerative diseases, cancer, and heart disease, with some promising results [6,7,8,9,10].

EGCG has also been shown to have antiviral activity against influenza viruses in vitro. Studies have demonstrated a half-maximal inhibitory concentration (IC_50_) between 5 and 30 µM against several influenza A virus strains, such as A/Puerto Rico/8/1934 (H1N1), A/Hong Kong/1968 (H3N2), and a recent A(H1N1)pdm virus A/Korea/1/09 [11,12,13,14,15,16]. In vitro activity has also been shown for severe-acute respiratory syndrome coronavirus 2 (SARS-CoV-2) [14,17,18,19,20], hepatitis C virus (HCV) [21,22], herpes simplex virus (HSV-1), vaccinia virus (VACV), adenovirus (AdV), reovirus (RV) [16], and others. Colpitts and Schang [16] have shown that EGCG interacts with several viruses (HCV, HSV-1, VACV, AdV, RV, and IAV) surface proteins and postulate that inhibition of influenza virus may be due to EGCG binding to the influenza surface glycoprotein haemagglutinin (HA) [16,18,23]. Another group demonstrated interactions between EGCG and influenza HA by crosslinking-mass spectrometry [23]. While these in vitro studies all cite an antiviral role, further research is needed to confirm the efficacy of EGCG in relevant in vivo models.

Small clinical trials have demonstrated the prophylactic benefits of EGCG against influenza and other upper respiratory tract infections [24]. However, a proportion of these studies were not randomised placebo-controlled trials (RCTs) and were classified by two meta-studies as having either ‘some concerns’ or a ‘high risk’ of bias [25,26]. The two blinded RCTs were conducted on health care workers in Japan, one of which demonstrated that drinking three cups of green tea per day (approximately 60 mg EGCG/day) reduced the incidence of upper respiratory tract infections from 28% (placebo, n = 86) to 13% (treated, n = 84) [27], and the other showed that six green tea extract capsules per day (270 mg EGCG/day) reduced the incidence of clinically defined influenza (fever and any two of the following symptoms; cough, sore throat, headache, or myalgia) from 13% (placebo, n = 99) to 4% (treated, n = 99) [28]. Two studies in mice demonstrated varying levels of antiviral efficacy for influenza A virus-infected animals, 10 or 40 mg/kg/day of oral EGCG treatment improved survival and reduced lung viral titres [15,29].

In vitro studies of EGCG against influenza (and other viruses, particularly SARS-CoV-2) are abundant, but this has not yet translated to any large cohort of human clinical trials; all previous studies have had fewer than 400 participants, and therefore uncertainty remains about the therapeutic benefits. Prior to human trials, robust in vivo studies are required, ideally against influenza virus strains that are currently circulating in the human population. Ferrets are considered the gold standard for influenza virus studies because they can be infected with human influenza viruses without any adaptation, and they display similar clinical signs and receptor distribution in the airways as humans [30].

Our study therefore aimed to explore the potential antiviral effect of EGCG (excluding other components of green tea extract) against human influenza A(H1N1)pdm09 viruses in vitro and in the ferret animal model, as well as the benefits of EGCG in combination with oseltamivir (the SOC for influenza treatment).

## 2. Materials and Methods

### 2.1. Cells

Madin–Darby Canine Kidney (MDCK CCL-34) cells (ATCC, Manassas, VA, USA) were cultured at 37 °C and 5% CO_2_ in Dulbecco’s Modified Eagle Medium (DMEM, high glucose pyruvate; Gibco, Waltham, MA, USA). DMEM was supplemented with 10% foetal bovine serum (FBS, Bovogen Biologicals, Australia), 1× GlutaMAX (Gibco, USA), 1× MEM non-essential amino acid solution (Gibco, USA), 0.05% sodium bicarbonate (Gibco, USA), 20 μM HEPES (Gibco, USA), and 100 U/mL penicillin-streptomycin solution (Gibco, USA). Maintenance medium (DMEM medium containing the above constituents excluding only serum) was used for virus dilutions. Maintenance medium supplemented with 2 µg/mL TPCK-treated trypsin (Infection medium; SAFC Biosciences, Lenexa, KS, USA) was used for virus infection protocols [3].

Calu-3 (HTB-55) cells (ATCC, USA) were cultured at 37 °C and 5% CO_2_ in Eagle’s Minimum Essential Medium (EMEM; Gibco, USA), supplemented with 10% FBS, 50 U/mL penicillin-streptomycin solution, 1× MEM non-essential amino acid solution, and 1 mM sodium pyruvate (Gibco, USA). During infection, Calu-3 medium was supplemented as above, although modified to include 0.3% FBS and 2 µg/mL TPCK-treated trypsin.

### 2.2. Compounds

(-)-Epigallocatechin-3-gallate (EGCG), C_22_H_18_O_11_, was produced with a purity of 98.95% by MedChemExpress (Monmouth Junction, NJ, USA). Stock solutions for in vitro work were diluted to 50 mM in dimethyl sulfoxide (DMSO; SAFC, USA) and stored at −80 °C. For in vivo studies, EGCG doses were prepared immediately prior to dosing by diluting EGCG in sugar water (1 g/mL sugar) to achieve a final concentration of 125 or 250 mg/mL. Tamiflu paediatric solution (Hoffman-La Roche, Basel, Switzerland), containing 0.5122 g of oseltamivir phosphate (OST) per bottle, was diluted to 5 mg/mL in sugar water.

### 2.3. EGCG Cytotoxicity Assays in MDCK and Calu-3 Cells

Confluent 96-well plates (Corning, Manassas, VA, USA) seeded with MDCK or Calu-3 cells were incubated at 35 °C in a 5% CO_2_ incubator for 24 h in the presence of two-fold serial dilutions of EGCG (concentration range of 1000 μM–4 μM). Negative control wells contained infection medium only. Cell viability was determined using the CellTiter-Glo^®^ Luminescent Cell Viability Assay as per the manufacturer’s instructions (Promega, San Luis, CA, USA), and luminescence was measured using a FLUOstar Optima luminometer (BMG Labtech, Cary, NC, USA). The EGCG concentration that reduces cell viability by 50% compared to the cell-only control (CC_50_) was calculated using non-linear regression analysis (GraphPad Prism 10, Boston, MA, USA). Each fit line had a goodness of fit greater than 0.90 R^2^.

### 2.4. Viruses

Two influenza A(H1N1)pdm09 virus strains, A/California/07/2009 and A/Victoria/2570/2009, were plaque purified, as previously outlined [31]. Virus stocks were prepared in MDCK cells and stored at −80 °C. The infectious virus titre was determined prior to use by a virus titration assay as described below and sequenced by Illumina iSeq 100 (the whole genome sequencing protocol outlined previously [31]). Sequence information are available on the GISAID database: A/California/07/2009 ID: EPI_ISL_31553 (although acquired HA-S200P and NP-G102R) and A/Victoria/2570/2009 ID: EPI_ISL_417210.

### 2.5. MDCK Cell-Based EGCG Viral Foci Reduction Assay

The susceptibility of influenza viruses to EGCG was determined using a cell-based assay described by Koszalka et al. [32], with modifications. Briefly, all viruses were initially titrated by half-log dilutions to 1000 focus-forming units (FFU)/well in the absence of drugs before proceeding with drug susceptibility. Confluent MDCK cells in 96-well plates were washed twice with PBS, and 50 µL of 2-fold serially diluted EGCG (100 to 0.4 µM final concentration) were added to the wells before the addition of 50 µL of virus diluted in infection medium. After incubation for 90 min at 35 °C, 5% CO_2_ 100 μL of overlay containing equal parts of 3.2% carboxymethyl cellulose (CMC) (1.6% final) (Sigma Aldrich, St Louis, MO, USA) and 2× MEM (1× final) (Sigma Aldrich, USA), supplemented with 2 μg/mL trypsin, were added to each well. The 2× DMEM was supplemented with 20 μM HEPES (Gibco, USA), 100 U/mL Penicillin-Streptomycin (Gibco, USA), and 0.06% Sodium Bicarbonate (Gibco, USA). Plates were incubated at 35 °C in a 5% CO_2_ gaseous incubator for 24 h before fixation with 10% formalin (Sigma Aldrich, USA) and permeabilisation with 0.5% Triton X-100 (Sigma Aldrich, USA). The cell monolayer was washed with wash buffer (0.05% Tween 20 (Sigma Aldrich) in PBS) and strained with influenza A nucleoprotein antibody MAB8251 (Millipore, Burlington, MA, USA) in 2% skim milk, followed by a goat anti-mouse IgG-horse radish peroxidase (Biorad, Hercules, CA, USA) and TrueBlue™ Peroxidase Substrate (KPL, Gaithersburg, MD, USA).

FFU were quantified using the Immunospot BioSpot 5.1.36 (CenturyLink Inc., Shaker Heights, OH, USA) as an average of duplicate wells. Using the mean FFU, the percentage inhibition of FFU was calculated using the following formula: Percentage inhibition = (100 − (X − CC)/(VC − CC)) × 100, where CC = FFU in cell control wells (no virus, no drug), VC = FFU in virus control wells (virus, no drug), and X = Mean FFU. Using the percent inhibition, the IC_50_ for EGCG of each virus was determined using non-linear regression analysis (GraphPad Prism, USA). Each fit line had a goodness of fit greater than 0.90 R^2^. Selective index (SI) was calculated as a quotient of the toxicity and IC_50_.

### 2.6. Calu-3 Cell-Based EGCG Virus Yield Reduction Assay

The susceptibility of influenza viruses to EGCG was determined in Calu-3 cells using a viral yield reduction assay, given that the foci reduction assay was not validated for Calu-3 cells. Cells were seeded 2–3 days prior to infection at 1.2 × 10^5^ cells/well in a 96-well plate. Confluent wells were washed with PBS, and the medium was replaced with infection medium containing EGCG at 2-fold dilutions (200 to 1.6 µM final concentration). A/California/07/2009 and A/Victoria/2570/2019, the virus was simultaneously added at a MOI of 0.01 (10^3.5^ TCID_50_/well) and incubated for 24 h at 35 °C in a 5% CO_2_ incubator before the virus-containing cell supernatant was stored for virus titration by the TCID_50_ assay (described below). The percentage of viral inhibition was quantified relative to the virus-only control titre. Using the percent inhibition, the IC_50_ for EGCG of each virus was determined using non-linear regression analysis (GraphPad Prism, USA). Each fit line had a goodness of fit greater than 0.92 R^2^. Selective index (SI) was calculated as a quotient of the toxicity and IC_50_.

### 2.7. Virus Titration Assay

Infectious virus titres were determined by a 50% tissue culture infective dose (TCID_50_) assay in MDCK cells, as previously described [33]. In brief, MDCK cells were seeded at 3.5 × 10^4^ cells/100 µL into a 96-well plate and cultured overnight at 35 °C in 5% CO_2_. The infectivity is determined by recording the presence of cytopathic effect (CPE) at four days post-infection when an inoculum of 20 µL of influenza virus, nasal washes, or tissue homogenates is applied in quadruplicate on a 96-well tissue culture of MDCK cells and is ten-fold serially diluted. The dilution at which 50% of the wells are infected is calculated using the Reed and Muench method [34].

### 2.8. Ferrets

Outbred male and female ferrets (Mustela putorius furo) were obtained from commercial breeders (Animalactic Animals & Animal Products Pty Ltd., Melbourne, Australia) and were a minimum of 12 weeks of age and 0.6 kg in body weight. Seronegativity against two recent A/(H1N1)pdm, A(H3N2) and B Victoria lineage human influenza virus strains (A/Victoria/2570/2019, A/Sydney/5/2021, A/Cambodia/e0826360/2020, A/Darwin/6/2021, B/Austria/1359417/2021, B/Phuket/3073/2013) was confirmed by haemagglutination inhibition assay. Ferrets were housed individually in high-efficiency particulate air-filtered cages with ab libitum food, water, and enrichment equipment throughout the experimental period. All animal procedures conducted in this study were approved by the University of Melbourne Animal Ethics Committee (project licence no. 20033) in accordance with the Australian Government, the National Health and Medical Research Council, and the Australian code of practise for the care and use of animals for scientific purposes (8th edition).

### 2.9. Ferret Toxicity Study

The ferret toxicity study included three groups of four ferrets treated with 1 mL/kg sugar water (placebo), 125 mg/kg EGCG, or 5 mg/kg paediatric oseltamivir treated twice daily (8 h apart) for 4 days. Treatment dosage was calculated as per individual ferret weight, delivered orally by syringe, and diluted in sugar water to encourage ingestion.

Blood was collected on D0, as well as D4 post-dosing commencing, and D14. Sera was sent for external analysis by Gribbles Veterinary Pathology, Melbourne (Australia). Serum biochemistry tests were performed to quantify: Albumin, ALP, ALT, Amylase, Anion gap, AST, Bicarbonate, Bilirubin (Total), Calcium, Cholesterol, Chloride, Creatinine, CK, CRP, GGT, Globulins, Glucose, Lipase, Magnesium, Phosphate, Potassium, Protein (Total), Sodium, and Urea.

### 2.10. Ferret Antiviral Treatment Study

The ferret drug efficacy study included four groups of four ferrets treated with 1 mL/kg sugar water (placebo), 250 mg/kg EGCG, 5 mg/kg paediatric oseltamivir, or their combination (dosage delivered as before). All ferrets received anaesthesia (1:1 (*v*/*v*) ketamine (100 mg/mL) and xylazine (20 mg/mL)) via intramuscular injection (IM), and 1 × 10^5^ TCID_50_ of influenza A(H1N1)pdm09 A/California/07/2009 virus in 500 µL of PBS were delivered by the intranasal route (250 µL per nostril). One ferret was found dead on day four post-infection, and a veterinary autopsy determined mortality by cardiovascular collapse.

Animals were treated twice daily (8 h apart) for 4 days, starting 1 h prior to intranasal infection, to align with our in vitro assays and maximise any treatment benefit of EGCG, as mechanism of action studies suggest an antiviral role in the early stages of influenza infection/replication. Also, previous in vivo studies have started EGCG treatment between 4 h prior and 2 h post-inoculation [15,29,35]. Twice daily treatment was performed to address the short plasma half-life of EGCG, demonstrated to be between 2 and 6 h in humans [36,37,38], and similar in rats, suggesting a more frequent dose may improve therapeutic benefit [39].

Ferrets were sedated daily (Xylazine; 5 mg/kg), monitored for clinical signs (subcutaneous microchip temperature and body weight), and nasal wash samples were collected with 1 mL of PBS. At day four post-inoculation, all animals were euthanised (Lethabarb; 0.5 mL/kg), and the five major lung lobes, nasal turbinates, and soft palate tissue were excised, weighed, and separately homogenised in PBS (10% *w*/*v* dilution for nasal turbinates and soft palates and 25% *w*/*v* dilution for lung lobes) using an Omni Tip^TM^ Homogenizer (PerkinElmer, Waltham, MA, USA). Residual cells and connective tissue were removed by twice centrifugation at 3000 rpm for 10 min. Tissue homogenate supernatants were stored at −80 °C. The infectious viral load of samples was determined by a virus titration assay.

### 2.11. Pyrosequencing

Viral RNA was extracted from a 140 µL ferret nasal wash or tissue homogenate aliquots using the QIAamp viral RNA mini kit (Qiagen, Melbourne, Australia). RT-PCR amplification of NA-275 gene regions was performed using biotin-tagged primers (Forward—GACAGGCCTCATACAAGATCTTC, Reverse—Biotin-TGCCAGTTATCCCTGCACACACA) and the MyTaq One-Step RT-PCR kit (Bioline, Sydney, Australia). The pyrosequencing analysis was performed on amplified cDNA with sequencing primers (AATGAATGCCCCTAATT) and the PyroMark Q96 ID system (Qiagen). Sequence analysis revealed the relative proportion of wild-type and SNPs at position NA-275, as previously outlined [40].

### 2.12. Statistics and Reproducibility

All statistical analysis performed throughout was named with the statistical test used and information about the exact sample size, any assumptions or corrections, and the resulting *p* value of the null-hypothesis tests. A standard deviation was used to capture errors from the mean in graphical representations. In terms of general reproducibility, in vitro studies were conducted with three or more independent experiments performed in triplicate, except for the yield reduction assay with five independent experiments performed with a single replicate (given 64 samples to be analysed by the TCID_50_ assay per experiment). In vivo, the antiviral treatment study was performed with four ferrets per group, and the toxicity study was performed with three ferrets per group.

## 3. Results

### 3.1. EGCG Demonstrated Antiviral Efficacy in MDCK and Calu-3 Cells

Using a cell-based drug susceptibility assay, we explored the efficacy of EGCG against two strains of influenza A(H1N1)pdm09 virus, A/California/07/2009 and the more contemporary A/Victoria/2570/2019, in Madin-Darby Canine Kidney (MDCK) and human lung carcinoma (Calu-3) cells. EGCG was active against both viruses, with a 50% inhibitory concentration (IC50) of 3.0 and 3.8 µM, respectively, in MDCK cells (Figure 1a) and 24 and 26 µM in Calu-3 cells (Figure 1c). This resulted in a selection index score (SI, 50% cytotoxic concentration (CC50)/IC50) of 96 µM for A/California/07/2009 and 77 µM for A/Victoria/2570/2019 in MDCK cells, given that the CC50 was 290 µM (Figure 1b). The CC50 in Calu-3 cells was 420 µM (Figure 1d), therefore the resulting SI was 18 and 16 for each virus, respectively, a 4 to 6-fold lower than the SI in MDCK cells.

### 3.2. EGCG Treatment in Uninfected Ferrets Reveals No Signs of Toxicity

Before EGCG antiviral studies in ferrets, we first determined the toxicity of 250 mg/kg/day oral EGCG dosage by measuring weight loss and blood biochemistry of uninfected ferrets, given that toxicity studies of EGCG in the literature had not previously included ferrets. Ferrets (n = 3 per group) were treated orally with EGCG, OST, or placebo twice daily for five days, and no significant effect on body weight or temperature was observed (Figure 2a,b). Blood sera biochemistry prior (d0), during (d4), and post-treatment (d14) revealed liver enzyme and substrate concentrations were within normal ranges except ALT levels for one ferret, which remained high before and after treatment (Figure 2c–h), as well as additional enzymes, substrates, and electrolytes (Appendix A), indicating healthy sera biochemistry in all EGCG or placebo-treated ferrets. Therefore, we deemed 250 mg/kg/day to be a safe dose of EGCG in the ferret model.

### 3.3. EGCG Shows No Reduction in Viral Titres of Influenza Infected Ferrets Compared to OST Treatment

To determine the antiviral efficacy of EGCG against influenza virus in vivo, oral treatment of ferrets commenced one hour prior to intranasal infection with A/California/07/2009, and animals were treated twice daily for four days (Figure 3a). A dosage of 500 mg/kg/day EGCG was used, given that no signs of toxicity were observed at 250 mg/kg/day, to maximise any antiviral effects observed. Daily nasal wash samples were collected from the ferrets over the course of the experiment, as well as tissue samples from the lungs, nasal turbinates, and soft palate. Infectious viral titres from daily nasal washes collected from four ferrets per group indicated no significant differences in EGCG-treated ferrets compared to placebo at all time points. Animals treated with OST and a combination of OST + EGCG had significantly reduced nasal wash viral titres at day 1 post-infection (Figure 3b). Area under the curve analysis confirmed no differences in the total viral shedding for any of the four treatment groups (Figure 3c). OST monotherapy significantly reduced mean pulmonary viral load by greater than 2 log_10_TCID_50_ compared to placebo. In contrast, treatment with EGCG alone had no significant reduction in lung viral titres. When OST and EGCG treatments were administered in combination, there was no difference in viral titre reduction compared to OST alone. Animals in all treatment groups observed no difference in viral loads in the nasal turbinate and soft palate tissues compared to placebo (Figure 3d,e). EGCG-only-treated ferrets also demonstrated significant weight loss three- and four-days post-infection compared to placebo, while OST-treated ferrets had reduced weight loss, although not significantly different from placebo (*p* = 0.14 at day 4) (Figure 3f). We also performed pyrosequencing to test for amino acid mutations that lead to oseltamivir antiviral resistance, in particular the mutation NA-H275Y, in nasal washes and tissue samples from each animal on day 4 post-infection. Only low levels of NA-H275Y were identified (<10% of the virus sample) (Appendix A).

## 4. Discussion

The efficacy of influenza antiviral drug therapy is threatened by the frequent emergence of antiviral-resistant influenza viruses; therefore, alternative treatment options such as polyphenol compounds are useful to explore for influenza treatment. EGCG, a lead candidate in several studies of polyphenol antiviral efficacy [11,42], led us to pursue this antiviral efficacy study. Our results indicate that EGCG has promising antiviral activity against influenza A(H1N1)pdm09 viruses in vitro; however, testing with the same A(H1N1)pdm09 influenza virus in the ferret challenge model showed that EGCG treatment was not able to reduce influenza viral load in the upper or lower respiratory tract and had no beneficial effect when combined with OST treatment.

In vitro cell-based drug susceptibility assays highlighted EGCG inhibition of virus replication but also indicated a disparity between the two cell lines utilised for these assays. MDCK cells demonstrated a lower IC_50_ value, resulting in a 4 to 6-fold increase in the SI compared to Calu-3 cells. Furubayashi et al. outlined differences in MDCK and Calu-3 tight junction integrity (measured by transepithelial electrical resistance) and transcellular drug absorption (measured by liquid chromatography) [43], which may differentially impact EGCG absorption. Discrepancies between these cell lines can impact influenza virus infection efficiency [44] as well as the uptake and localisation of exogenous compounds [45], underscoring the need to test antiviral efficacy in more than one relevant cell line.

Our in vivo results differ from previous EGCG oral treatment studies in mice. Ling et al. showed that treatment with 40 mg/kg/day of EGCG significantly reduced viral lung titres and mortality following a lethal dose of an influenza mouse-adapted virus (A/FM/1/47), comparable to the observed antiviral effects of OST. However, at 10 mg/kg/day of EGCG, this effect was no longer observed [15]. Another mouse study demonstrated that EGCG treatment with 10 mg/kg/day reduced mortality, weight loss, lung damage, and lung viral titres after swine influenza A(H9N2) virus infection (A/swine/HeBei/012/2008) [29]. Lee et al., however, could not repeat these results with green tea extract (including EGCG) treatment of mice infected with an influenza mouse-adapted virus (A/NWS/33), as there was no reduction in mortality at 1, 10, or 100 mg/kg/day; however, a significant reduction in lung viral titres was observed at the highest concentration [35]. These studies failed to determine a consistent effective concentration of EGCG treatment in mice, although likely greater than 10 mg/kg/day. Our study with a human influenza A(H1N1)pdm09 virus infection in ferrets, the gold-standard animal model for influenza, had a much higher dose (500 mg/kg/day), twice daily dosing (unlike once daily for the above mouse studies), and still did not yield an observed antiviral effect. However, ferret pharmacokinetics studies are necessary to confirm plasma and tissue EGCG levels.

The ferret antiviral efficacy study showed that OST treatment significantly reduced viral pulmonary burden but had no effect on the viral titres in the upper respiratory tract (nasal turbinates and soft palate) compared to placebo, which closely aligns with previous antiviral studies in ferrets [46,47,48]. EGCG alone had no effect on viral titers. In addition, combination therapy (EGCG + OST) had no effect on viral titres compared to OST alone treated ferrets, indicating that EGCG is unlikely to have additive benefits with OST in vivo. Thus, given the efficacious results of OST treatment, the early timing of EGCG treatment (immediately prior to infection), and the high EGCG dose relative to other animal studies, it is unlikely that EGCG has an antiviral effect against influenza A virus infection in ferrets.

EGCG has demonstrated promising antiviral properties in vitro, although there have been mixed reports of its efficacy in animal models. Previous conclusions about antiviral efficacy from poorly designed in vivo or purely in vitro studies may misrepresent relevant clinical utility. It may be especially important for naturally occurring antiviral compounds to be supported by gold-standard animal studies; for example, Vitamin D (calcitriol) showed strong inhibition of SARS-CoV-2 in vitro but no effect in mice [49]. Several factors may explain the inconsistency between in vivo studies, such as bioavailability in different animals, compound instability (rapid oxidation), and inactivation in the gut [50]. EGCG bioavailability and stability are partially improved when mixed with salmon oil and ascorbic acid [50,51]. Another improvement may include a different route of drug administration; for example, black tea extracts (theaflavin polyphenols) were shown not to be well absorbed by oral dosing and may be better administered by intraperitoneal route [52,53]. Nasal vapour delivery may also bypass the inactivation of EGCG in the gut, improving EGCG bioavailability at the site of infection. Aree and Jongrungruangchok showed that EGCG had improved stability and anti-oxidant properties when combined with cyclodextrin [54]; these findings were recently utilised to demonstrate the antiviral efficacy of the EGCG-cyclodextrin mixture against several influenza viruses in vitro [14]. Interestingly, mouse models used to date have demonstrated the antiviral efficacy of oral EGCG in the absence of drug-stabilising compounds [15,29,35]. Further studies may explore plasma and mucosal tissue concentrations of EGCG post-oral, peritoneal, or nasal vapour delivery, as well as different stabilising additives such as cyclodextrin to improve bioavailability.

EGCG is a compound metabolised by the liver [55], and toxicity studies have demonstrated some liver toxicity, which may potentially counteract any antiviral effects. Two 14-week dosing studies observed no adverse effect level (NOAEL) for the liver in mice, rats, and non-fasted dogs when 500 mg/kg/day of EGCG was administered. However, treatment-related mortality likely related to liver necrosis was observed at 1000 or more mg/kg/day [56,57]. No toxicity studies have been performed on ferrets until now. We have found that 250 mg/kg/day in ferrets did not result in changes to liver function, according to enzyme and substrate homeostasis, nor did it result in significant weight loss. However, at the higher dose of 500 mg/kg/day (increased from 250 mg/kg/day to maximise the potential antiviral effect), influenza-infected ferrets treated with EGCG had higher weight losses than placebo-treated animals, indicating that at this higher dose level, EGCG may have had some adverse effects in ferrets. It is also worth noting that weight loss as an indicator of disease/toxicity severity in ferrets is difficult to conclusively analyse given the genetic variability between animals [58,59].

A limitation of this study was to focus on the influenza A(H1N1)pdm09 subtype alone. Experiments performed with contemporary A(H3N2) and B influenza viruses would improve our understanding of EGCG in vitro activity. A(H3N2) and B viruses do not replicate well in the lungs of ferrets; therefore, antiviral efficacy studies in vivo would be difficult to assess and relate to in vitro findings.

The promising antiviral effect observed in vitro for influenza and other viruses may present opportunities for alternative uses for EGCG outside of antiviral therapy. EGCG has also shown potential as an adjuvant for influenza vaccines, improving vaccine immunogenicity in combination with aluminium salts [23,60], as well as dietary EGCG improving influenza vaccine response [61].

Although our study demonstrated the in vitro antiviral efficacy of EGCG against the influenza A(H1N1)pdm09 viruses tested, the in vivo results obtained in this study do not support EGCG as an effective anti-influenza therapeutic drug in the conditions tested in this study. However, further studies in ferrets should include different EGCG dosages and routes of administration, as well as compounds to support EGCG stability and bioavailability, as these factors may have contributed to poor antiviral efficacy.

## Figures and Tables

**Figure 1 viruses-15-02447-f001:**
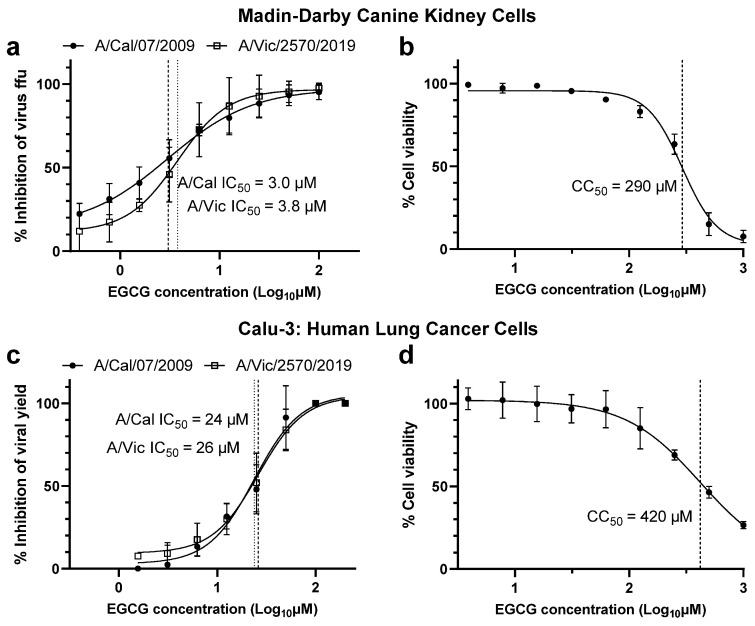
EGCG influenza virus inhibition and cell toxicity in vitro. Cells in a 96-well plate were treated with EGCG at 2-fold dilutions for 24 h. (**a**) MDCK cells infected with A/California/07/2009 and A/Victoria/2570/2019 were quantified for virus focus forming unit (FFU) relative to the infected but not treated control (n = 4, in triplicate). (**b**) MDCK cell viability was determined by the CellTiter-Glo^®^ Luminescent Cell Viability Assay (n = 3, in triplicate). (**c**) Yield reduction assay was conducted using Calu-3 cells to determine the relative inhibition of EGCG against A/California/07/2009 and A/Victoria/2570/2019 (n = 5, in single). (**d**) Calu-3 cell viability was also determined by the CellTiter-Glo^®^ Luminescent Cell Viability Assay (n = 5, in triplicate). Each point represents the mean and standard deviation. IC50 or CC50 is indicated on each graph by vertical dashed lines.

**Figure 2 viruses-15-02447-f002:**
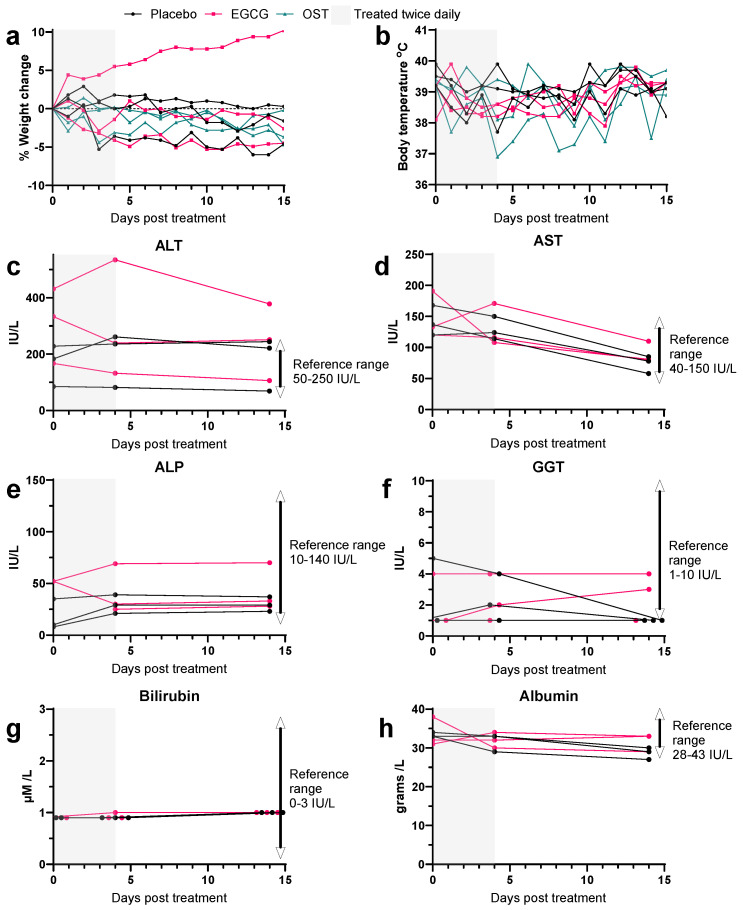
Weight loss, temperature, and liver enzyme/substrate analysis of uninfected ferrets treated with EGCG. Ferrets were treated with 1 mL/kg of sugar water (placebo) or 125 mg/kg EGCG per dose twice daily (8 h apart) for 5 days (d0–d4) or 5 mg/kg paediatric OST per dose twice daily (8 h apart) for 5 days (d0–d4). (**a**) The weight change of animals weighed daily is shown by solid lines. (**b**) Body temperature was recorded daily on a microchip, as shown by solid lines. For liver enzyme analysis, blood was harvested at d0, d4, and d14 post-treatment onset for biochemistry analysis of sera. Liver/biliary tract enzymes/substrates indicative of liver function, including (**c**) alanine transaminase (ALT), (**d**) aspartate transaminase (AST), (**e**) alkaline phosphatase (Alk Phos), (**f**) gamma-glutamyl transferase (GGT), (**g**) total bilirubin, and (**h**) albumin, were quantified with a reference range sourced from Hein et al. [41]. Three ferrets were used in each group.

**Figure 3 viruses-15-02447-f003:**
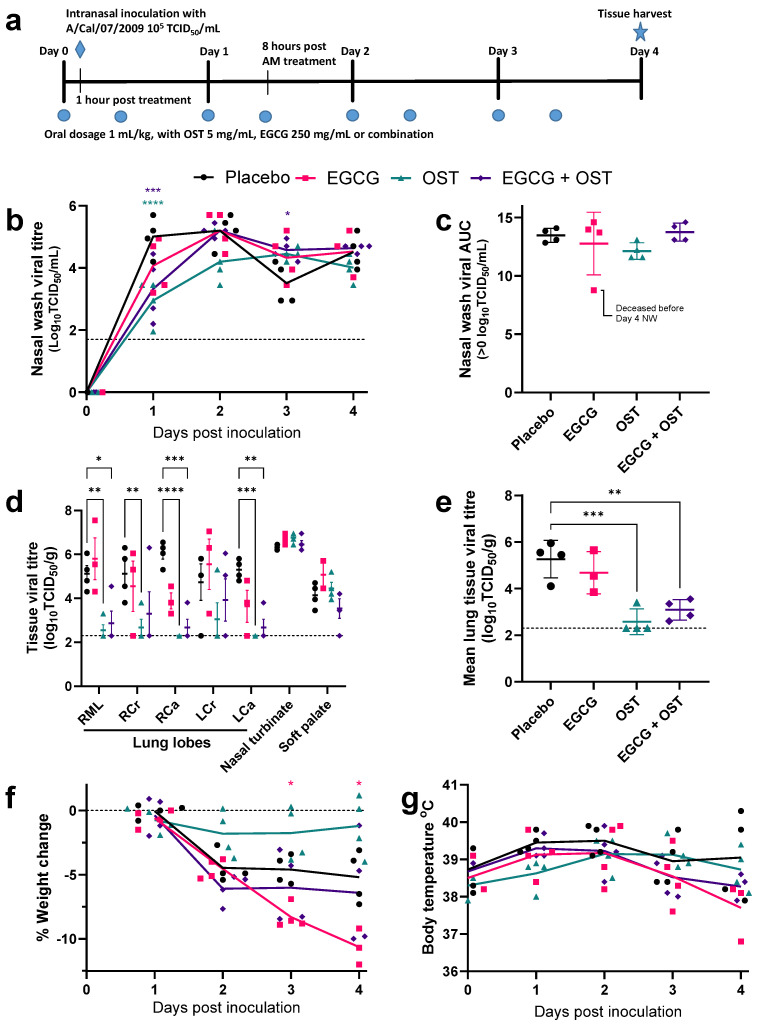
EGCG and Oseltamivir combination therapy in ferrets infected with A/California/07/2009 (**a**) Ferrets were treated with 1 mL/kg of sugar water (placebo), 250 mg/kg of EGCG, 5 mg/kg of paediatric oseltamivir (OST), or their combination twice daily (8 h apart) for 4 days, starting 1 h prior to intranasal inoculation with 5 log_10_TCID_50_/500 μL with A/California/07/2009. (**b**) Daily nasal wash samples were analysed by the TCID_50_ assay, with the mean titre indicated by the lines. The limit of detection (LOD) is indicated by the dashed line at 1.7 log_10_TCID_50_/mL. (**c**) The mean area under the curve (AUC) was determined from the shedding curves of individual ferrets over time. (**d**) At day four post-infection, respiratory tract tissues were collected for a virus titration assay. (**e**) The mean titre for all five major lung lobes shown for each animal. LOD is indicated by the dashed line at 2.3 log_10_TCID_50_/g. (**f**) Weight loss was determined daily relative to day 0 starting weight, with the mean percentage weight change indicated by the lines. (**g**) Body temperature was recorded daily from a subcutaneous microchip, with the mean temperature indicated by the lines. Data from four ferrets determined the mean and SD, compared across each group by Two-way ANOVA. Tukey’s multiple comparisons (* *p* < 0.05, ** *p* < 0.01, *** *p* < 0.001, **** *p* < 0.0001), or ordinary one-way ANOVA Tukey multiple comparisons for (**b**,**d**). Data representing the EGCG-treated group at day four are from only three ferrets due to mortality.

## Data Availability

The data presented in this study are included in this published article and its Appendix A. Virus sequence information can be found here: gisaid.org.

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
