# Peer review of "Pre-Clinical Evaluation of the Antiviral Activity of Epigalocatechin-3-Gallate, a Component of Green Tea, against Influenza A(H1N1)pdm Viruses"

_viruses, 2023, doi:10.3390/v15122447_

Round 1
Reviewer 1 Report
Comments and Suggestions for Authors
In the study, researchers investigated the potential antiviral efficacy of epigallocatechin-3-gallate (EGCG), a polyphenol found in plants, against influenza. Prior research had shown favorable results in vitro and in mouse studies, but the current study aimed to assess its effectiveness using the ferret model. The study concluded that although EGCG exhibited antiviral activity against influenza viruses in vitro, it did not effectively hinder viral replication in the respiratory tract of ferrets. This suggests that while polyphenols like EGCG, their effectiveness can vary across different experimental models.
Did the authors perform any bioavailability test of EGCG?
Fig 3f: EGCG treatment alone results in significant weight loss of ferrets as compared to placebo. Do the authors think that high dosage of EGCG given to ferrets(500mg/kg/day) is contributing to toxicity and resulting in no change in antiviral activity?
Reviewer 2 Report
Comments and Suggestions for Authors
In their article” Pre-clinical evaluation of the antiviral activity of Epigalocatechin-3-gallate, a component of green tea, against influenza A(H1N1)pdm viruses “ the authors present a study, whose aim is to explore the antiviral efficacy of polyphenol epigallocatechin-3-gallate in vitro and in ferrets.
The information in the manuscript is interesting, first because influenza viruses are still of a big concern for human health, and second because the development and investigation of new antiviral drugs is significant. Most Influenza antiviral drugs have resistance mutations and it is relevant to search for new mechanisms for successful treatment.
I have some minor points for the author to consider:
1. In the first paragraph lines 33-36 there are repetitions of the word “burden”
2. Materials and Methods.
· Cells. At the end of the first paragraph, you have to cite.
· Viruses. Why did you choose exactly these two viruses?
· MDCK cell-based EGCG viral foci reduction assay. The proper citation in this paragraph is: Koszalka P., Tilmanis D., Roe M., Vijaykrishna D., Hurt A. C. Baloxavir marboxil susceptibility of influenza viruses from Asia-Pasific, 2012-2018.Antiviral Research, Volume 164, April 2019, Pages 91-96.
· Ferret antiviral treatment study. Lines 224-225. Ferret from which group was found dead?
3. Results. Lines 265-268. You do not need this paragraph.
Reviewer 3 Report
Comments and Suggestions for Authors
Comments for the author of Viruses manuscript viruses-2712767:
The author of the Viruses manuscript “Pre-clinical evaluation of the antiviral activity of Epigalocatechin-3-gallate, a component of green tea, against influenza A(H1N1)pdm viruses”, present findings from their research testing the polyphenol epigallocatechin-3-gallate (EGCG) as an antiviral against influenza. Specifically, this team is building on previous research testing EGCG in cell and mouse models to test efficacy in ferrets. To begin these studies, they test EGCG in Madin-Darby Canine Kidney (MDCK) and human lug carcinoma (Calu-3) cells. Their results show antiviral activity for EGCG in both cell lines using both the A/California/7/2009 and A/Victoria/2570/2019 H1N1pdm09 influenza viruses. When tested in ferrets, using a dose of 500 mg/kg/day for 4 days, there was no change in respiratory tissue virus titers. This test under-performed when compared with oseltamivir, and the authors conclude that ECGC has antiviral activity in cells but not in ferrets. Below are some comments that I would like the authors to address as they revise the manuscript.
General Comments:
- It is concerning that one ferret died during inoculation. Is there any potential thoughts on what caused the cardiovascular collapse? Was this work done in accordance with approved animal protocols, and have these techniques been used for ferret inoculations in the past? I have experience with the ferret model of influenza and am surprised to see this outcome from a virus inoculation (although we use isoflurane for inoculations and ketamine only for nasal wash collections). It would be good to indicate whether this has led to a change in future practice and/or what a veterinarian had to say about this outcome.
- The first paragraph in the Results section (lines 266-268) appear to be instructions from the journal and not part of the actual manuscript text.
- Can the authors explain the large error bars in Figure 2a (EGCG group). There is a noticeable difference between this group and the other two, and it might help to know if there was an outlier that skewed the EGCG results. Was this the same ferret with the elevated ALT levels?
- Why did you choose to use a higher dose of EGCG (500 mg/kg/day) than was tested in the toxicity study (250 mg/kg/day)? It is mentioned to maximise antiviral effects, but why then didn’t you test 500 mg/kg/day in the toxicity study?
- Why was the deceased ferret counted in the nasal wash titer? It says deceased before Day 4, but it was implied in the Methods section that the ferret died at inoculation. Can we please have a more thorough description of what happened with this ferret? And the input of a veterinarian?
- With the information available regarding EGCG bioavailability, including salmon oil and ascorbic acid, delivery route, and cyclodextrin, I wonder if the authors could comment on why they chose the delivery method that they used in this study. Specifically, what method was used to define of route and method of delivery.
Round 2
Reviewer 3 Report
Comments and Suggestions for Authors
Thank you for clarifying the previous concerns and addressing all comments.